# Immune Transcriptome and Secretome Differ between Human CD71^+^ Erythroid Cells from Adult Bone Marrow and Fetal Liver Parenchyma

**DOI:** 10.3390/genes13081333

**Published:** 2022-07-26

**Authors:** Roman Perik-Zavodskii, Olga Perik-Zavodskaya, Yulia Shevchenko, Vera Denisova, Kirill Nazarov, Irina Obleuhova, Konstantin Zaitsev, Sergey Sennikov

**Affiliations:** 1Laboratory of Molecular Immunology, Federal State Budgetary Scientific Institution, Research Institute of Fundamental and Clinical Immunology, Yadrincevskaya 14, 630099 Novosibirsk, Russia; zavodskii.1448@gmail.com (R.P.-Z.); okoneva94@gmail.com (O.P.-Z.); shevcen@ngs.ru (Y.S.); kirill.lacrimator@mail.ru (K.N.); obleukhova.irina@yandex.ru (I.O.); 2Clinic of Immunopathology, Federal State Budgetary Scientific Institution, Research Institute of Fundamental and Clinical Immunology, Zalesskogo 2/1, 630047 Novosibirsk, Russia; verden@bk.ru; 3Federal State Budgetary Scientific Institution “Siberian Federal Research and Clinical Center of the Federal Medicobiological Agency”, Rozy Lyuksemburg 5, 634009 Tomsk, Russia; limdff@yandex.ru

**Keywords:** CD71^+^ erythroid cells, bone marrow, fetal liver parenchyma, MIF, IL8, TLR1, TLR2, TLR9, NOD2

## Abstract

CD71^+^ erythroid cells (CECs) were only known as erythrocyte progenitors not so long ago. In present times, however, they have been shown to be active players in immune regulation, especially in immunosuppression by the means of ROS, arginase-1 and arginase-2 production. Here, we uncover organ-of-origin differences in cytokine gene expression using NanoString and protein production using Bio-Plex between CECs from healthy human adult bone marrow and from human fetal liver parenchyma. Namely, healthy human adult bone marrow CECs both expressed and produced IFN-a, IL-1b, IL-8, IL-18 and MIF mRNA and protein, while human fetal liver parenchyma CECs expressed and produced IFN-a, IL15, IL18 and TNF-b mRNA and protein. We also detected TLR2 and TLR9 gene expression in both varieties of CECs and TLR1 and NOD2 gene expression in human fetal liver parenchyma CECs only. These observations suggest that there might be undiscovered roles in immune response for CECs.

## 1. Introduction

CD71^+^ erythroid cells (CECs) are erythrocyte precursors that are known to be involved in immune regulation, especially immunosuppression. They establish immunosuppression through the means of ROS [1], arginase-1 [2], arginase-2 [3,4,5] and TGF-β [6] production and the presence of PDL-1 on their cellular surface [5]. CECs received their name due to the fact that they bear receptors for holotransferrin (CD71). CD71’s expression starts at the BFU-E stage and ends at the orthochromatic erythroblast stage of the erythroid cells’ differentiation [7]. CD71 may also be present on other cell types, such as activated T-cells [8]. CECs from both humans and mice have been shown to be cells with cytokine gene expression and cytokine protein production properties [6,9,10,11,12,13]. CECs are present at different stages of ontogenesis. CECs are found in the yolk sac [14], the fetal liver parenchyma, the fetal thymus and the bone marrow [15], as well as in the umbilical cord blood [16] during intrauterine development. In adults, CECs are typically found in the bone marrow [17], but cases of extra-medullary hemopoiesis in the liver, the spleen and the lymph nodes in case of pathology are also possible [18]. CECs were also shown to be quite abundant and associated with bad prognosis in the course of important diseases such as COVID-19 [19] and during sepsis [20]. In this work, we validate previous findings of cytokine gene expression and production and assess differences between CECs from the human fetal liver parenchyma and human adult bone marrow.

## 2. Materials and Methods

### 2.1. Study Population

Healthy bone marrow donors were 23–26 years old and were evenly distributed by sex (total: *n* = 6; males: *n* = 3; females: *n* = 3). Fetal liver parenchyma tissue samples were extracted after an abortion at 20–22 weeks of pregnancy; their sex was unknown (*n* = 6).

### 2.2. Study Interventions

We performed punctures and bone marrow aspirations at either the sternum (*n* = 1) or the ilium (*n* = 5).

### 2.3. Participant Safety

Three days before the bone marrow harvesting procedure, the donors underwent medical testing involving complete blood count, ECG and qPCR for COVID-19. We monitored the donors for 60 min after the bone marrow harvesting procedure.

### 2.4. Cell Isolation

We collected the bone marrow aspirates (up to 5 mL in volume) into tubes containing EDTA. We thawed fetal liver homogenized parenchyma cell samples stored in 10% DMSO and 90% FBS (up to 1.5 mL in volume) in a water bath at 37 °C, then washed them with 6 mL of a mixture containing 5 mL full RPMI 1640 cell culture medium and 1 mL FBS. We isolated bone marrow and fetal liver parenchyma mononuclear cells using density gradient centrifugation (Ficoll-Paque^TM^ (Thermo Fisher Scientific, Waltham, MA, USA) with a density of 1.077 g/mL) at 266 RCF for 30 min in order to remove RBCs and reticulocytes.

### 2.5. Magnetic Separation

We performed magnetic separation of both the bone marrow and fetal liver parenchyma mononuclear cells using a magnetic stand, a magnet (Miltenyi Biotec, 130-042-102, Bergisch Gladbach, Cologne, Germany) and CD71 MicroBeads (Miltenyi Biotec, 130-046-201, Bergisch Gladbach, Cologne, Germany) according to the manufacturer’s protocols.

### 2.6. Viability Staining

We measured the magnetically sorted cells’ viability on a Countess 3 Automated Cell Counter (Thermo Fisher Scientific, Waltham, MA, USA) according to the manufacturer’s protocols using trypan blue. Trypan blue staining showed >95% viability for both adult bone marrow and fetal liver parenchyma CECs.

### 2.7. Flow Cytometry

We washed the magnetically sorted cells in PBS. The following antibodies were used for staining according to the manufacturer’s protocols: anti-CD71-PE (BioLegend, 334106, San Diego, CA, USA) and anti-CD235a-FITC (BioLegend, 349104, San Diego, CA, USA). Flow cytometry showed >94% purity of the cells. The gating strategy was to isolate cells from debris, isolate singlets from the cells and measure CD71 in the singlets for both adult bone marrow and fetal liver parenchyma CECs (Figure 1a,b).

### 2.8. Cell Culturing

We cultured the magnetically sorted cells in X-VIVO 10 serum-free medium (Lonza, Basel, Switzerland) with the addition of x1 Insulin-Transferrin for 24 h at a concentration of 1 million per mL of the medium in order to support their viability and measure the culture medium’s cytokines afterwards.

### 2.9. Total RNA Extraction

We isolated total RNA from CECs after their magnetic separation and before culturing using a Total RNA Purification Plus Kit (Norgen Biotek, 48400, Thorold, ON, Canada), measured concentration of the RNA on a NanoDrop 2000c (Thermo Fisher Scientific, Waltham, MA, USA) and diluted the RNA to a concentration of 10 ng/μL using nuclease-free water. We froze the diluted total RNA at −80 °C until the immune transcriptome profiling.

### 2.10. Cell Culture Medium Harvesting

We separated the cell culture medium from the cells after the 24 h of culturing. We performed the separation by centrifugation at 1500 rpm for 10 min, transferred the cell culture medium into new 1.5 mL tubes with the addition of BSA up to a total concentration of 0.5% and froze the cell culture medium at −80 °C until the cytokine quantification.

### 2.11. Immune Transcriptome Profiling

We used 100 ng of the total RNA for each sample (*n* = 4, for both adult bone marrow and fetal liver parenchyma CECs) to profile 579 immunity-related genes in CECs. We hybridized the RNA to a NanoString Immunology v2 panel (NanoString, XT-CSO-HIM2-12, Seattle, WA, USA) according to the manufacturer’s protocols and analyzed the RNA on the NanoString Sprint instrument (NanoString, Seattle, WA, USA).

### 2.12. Cytokine Quantification in Culture Medium

We used 50 μL of adult bone marrow CECs’ (*n* = 6) and fetal liver parenchyma CECs’ (*n* = 6) culture media to quantify culture medium cytokines in doubles using a Bio-Plex Pro Human Cytokine Screening Panel, 48-Plex (BioRad, #12007283, Hercules, CA, USA) and a Bio-Plex 200 instrument (BioRad, Hercules, CA, USA).

### 2.13. Data Analysis

We analyzed the profiled CECs’ immune transcriptome data using the nSolver 4 software. At first, we performed background thresholding using the mean of the negative controls plus 2 standard deviations on the raw data counts in order to filter out >95% of the noise, and we normalized the data using spiked-in synthetic positive controls and the ALAS1, PPIA, POLR1B and SDHA housekeeping genes. We then cleaned out the noise values. We identified the noise values as the lowest count detected for each sample. We compared cytokine quantities in CECs’ culture media using multiple Mann–Whitney U-tests with Q = 0.01 in the GraphPad Prism 9 software.

## 3. Results

### 3.1. Cytokine Gene Expression Profiles of the CECs

We analyzed the CECs’ transcriptome profiles and found mRNAs of several cytokines. Detected cytokine mRNAs’ data are presented in Table 1.

### 3.2. Cytokine Protein Profiles in CECs’ Culture Medium

We found that both varieties of CECs had cytokine proteins in their culture medium. Both adult bone marrow and fetal liver parenchyma CECs’ culture media contained IFN-a and MIF. Only adult bone marrow CECs’ culture medium contained IL-1B and IL-8. Only fetal liver parenchyma CECs’ culture medium contained IL-15, IL-18, TNF-a and TNF-b. Cytokine concentration data and the presence of the corresponding mRNAs are presented in Table 2.

We observed that IL-8 and MIF were reliably more abundant in adult bone marrow CECs’ culture medium (Figure 1). Cytokines were considered reliably more abundant if the mean difference was >50 pg/mL and the *q*-value was <0.001.

### 3.3. CECs Have TLR Gene Expression

We found that there was pattern-recognition receptor gene expression in both adult bone marrow and fetal liver parenchyma CECs (Table 3).

## 4. Discussion

In the present study, we investigated the transcriptome and secretome profiles of human adult bone marrow CECs from donors from 23 to 26 years of age and human fetal liver parenchyma CECs.

We found transcripts of a plethora of cytokines. We also found corresponding cytokine proteins in CECs’ cell culture media. With that data in mind, we can say that: (1) both varieties of CECs produce IFN-a and MIF; (2) adult bone marrow CECs can additionally produce IL-1b and IL-8; (3) fetal liver parenchyma CECs can additionally produce IL-15, IL-18, TNF-a and TNF-b.

IL-8 and MIF were also reliably more abundant in adult bone marrow CECs’ culture medium. It is known that IL-8 [21,22,23] and MIF [24,25,26] direct cells to the bone marrow and promote cell growth and angiogenesis. We propose that adult bone marrow CECs can be involved in these biological processes through the means of secreting such cytokines.

Cytokines, found in fetal liver parenchyma CECs’ culture medium, could be described as pro-inflammatory and often involved in auto-immune processes [27,28]. However, it might be that their effects in vivo are masked by CECs’ overall induced immunosuppression.

We also detected gene expression of pattern-recognition receptors in both varieties of CECs. This can mean that CECs from both sources can recognize PAMPs through TLR9 [29], and human fetal liver parenchyma CECs can recognize PAMPs through TLR1/TLR2 [30] and NOD2 [31]. Pattern-recognition receptors were previously found in CECs of mice (TLRs 1, TLR2 and TLR4) [6], fish (TLR3, TLR9 and TLR21) [32,33] and birds (TLR2, TLR 3, TLR 4, TLR 5 and TLR21) [33,34], so it is quite important to test CECs for responses to various TLR ligands.

## 5. Conclusions

Overall, we observed that adult bone marrow and fetal liver parenchyma CECs are two heterogeneous cellular populations with different immune transcriptomes and somewhat different cytokine production profiles, possibly with different functions.

## Data Availability

The NanoString gene expression data were deposited to the Gene Expression Omnibus (GEO) with the accession code GSE199228. Other data that support our findings will be available from the authors by correspondence upon request.

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
