# Peer review of "Immune Transcriptome and Secretome Differ between Human CD71+ Erythroid Cells from Adult Bone Marrow and Fetal Liver Parenchyma"

_genes, 2022, doi:10.3390/genes13081333_

Round 1

Reviewer 1 Report

The authors describe an investigation into the the function of CECs. They advance our understanding of these newly discovered cells by showing which chemokines are expressed and secreted by cells, and showing this is context specific. They show this by a well designed experiment: they isolated CECs from fetal lever and bone marrow aspirates and quantified their expression and secretion of chemokines.

Major problems:

1. The main drawback of their work is the small number of samples: it is hard to prove generality with such a small number of observations.

2. The conclusions about FL parenchyma in lines 150-152 are not supported by the findings.

3. The whole discussion of the role of TLRs (lines 156-164) is not supported by the reported findings.

Minor problems:

4. Table 1 (in the results section, not named in the work) is not clear. Please add title and legend.

5. Very long sentences.

6. Commas where periods should be used.

Author Response

Major problems:

  1. The main drawback of their work is the small number of samples: it is hard to prove generality with such a small number of observations.

Sadly, these samples are extremely rare to find due to their fetal origin and we used our full stock of biological replicates in this study. The cell bank from which we acquired the fetal liver parenchyma samples has run out of stock, and no other local cell bank has such a tissue at the moment as well.

  1. The conclusions about FL ‪parenchyma in lines 150-152 are not supported by the findings.

Removed.

  1. The whole discussion of the role of TLRs (lines 156-164) is not supported by the reported findings.

This paragraph got lost during the pre-processing of the article. We restored it.

Minor problems:

  1. Table 1 (in the results section, not named in the work) is not clear. Please add title and legend.

This legend got lost during the pre-processing of the article. We restored it.

  1. Very long sentences.

Either, shortened, clarified or re-written.

  1. Commas where periods should be used.

Fixed.

Reviewer 2 Report

The authors present a very short work by NanoString technology to reveal organ-of-origin differences in cytokine gene expression between CECs from healthy adult bone marrow and CECs from fetal liver parenchyma. Generally, this study is well designed and simply analyzed, it was recommend to reorganize the manuscript into the type of communication or brief report instead.

1)      In Table 1, please clarify the definitions of ABM and FL.

2)      Line 48  Please consider deleting the indefinite article an.

3)      Line 67  Change mixture to “a mixture” or “the mixture”.

4)      Line 85  Change gating to “the gating”.

5)      Line 151  Change abundancy to abundance.

6)      Line 155  Proposed deletion of the indefinite article “a” before “more”.

Author Response

1)      In Table 1, please clarify the definitions of ABM and FL.

Clarified.

2)      Line 48  Please consider deleting the indefinite article an.

Deleted.

3)      Line 67  Change mixture to “a mixture” or “the mixture”.

Changed.

4)      Line 85  Change gating to “the gating”.

Changed.

5)      Line 151  Change abundancy to abundance.

Changed.

6)      Line 155  Proposed deletion of the indefinite article “a” before “more”.

Deleted.

Round 2

Author Response

Moderate English changes were made. We bolstered every lacking field in our manuscript.

Round 3
